# The Potential for Tidal Range Energy Systems to Provide Continuous Power: A UK Case Study

**Lucas Mackie** [1,*] , **Daniel Coles** [2] , **Matthew Piggott** [1] **and Athanasios Angeloudis** [3]

1   Department of Earth Science & Engineering, Imperial College London, London SW7 2AZ, UK;
    m.d.piggott@imperial.ac.uk
2   School of Engineering, Computing and Mathematics, Faculty of Science and Engineering,
    University of Plymouth, Plymouth PL4 8AA, UK; daniel.coles@plymouth.ac.uk
3   School of Engineering, Institute for Infrastructure & the Environment, The University of Edinburgh,
    Edinburgh EH8 9YL, UK; A.Angeloudis@ed.ac.uk
*   Correspondence: l.mackie18@imperial.ac.uk; Tel.: +44-7599-521-263

**Abstract:** The extraction of tidal energy from head differences represents a predictable and flexible option for generating electricity. Here, we investigate the generation potential of prospective tidal power plants in the UK. Originally conceived as separate projects, operating these schemes as a cooperative system could prove beneficial. Combined with the inherent operational flexibility of tidal range-based schemes, a notable tidal phase difference in selected sites allows for the system to spread power generation over a larger proportion of the day. Using depth-averaged modelling and gradient-based optimisation techniques, we explore how a flexible cumulative operation schedule could be applied to provide a degree of continuous supply if desirable. While fully continuous operation is not achieved, a number of different optimisation schedules deliver cumulative continuous supply for over half of the year. The average minimum cumulative power output on these days is consistently over 500 MW out of a total installed capacity of 6195.3 MW. Furthermore, by introducing financial incentives associated with reliable, baseload supply, we provide an economic assessment of the tidal power plant system. The daily minimum cumulative power output determines income in the modelled idealised baseload market, while excess supply is traded in an hourly variable wholesale energy market. Results indicate that subsidies would be required in order to make a pursuit of continuous generation financially advantageous over energy maximisation strategies.

**Keywords:** tidal range energy; resource variability; energy extraction; optimisation; baseload demand; flexible operation; numerical modelling

## 1. Introduction

Tidal power plants defer the ebb and/or flood of the tide by an impoundment. This creates an artificial basin and a head difference from which potential energy can be converted to electricity via a system of turbines [1]. The temporal variability of the tidal range resource [2] results in power generation intermittency in single-basin schemes [3]. However, where tidal range and stream energy technologies stand out over other intermittent renewable energy sources (e.g., wind energy) is the reliability and predictability of their resource [1]. Water elevation and velocity can be predicted at a given location through harmonic reconstruction of previously analysed elevation- and velocity-time series data [4], with further precision provided via numerical modelling [5]. An advantageous aspect specific to tidal range schemes in the renewable energy sector is the potential for operational flexibility [6]. The short-term storage capabilities possessed by tidal power plants is facilitated by the degree of control in scheduling different modes of operation. Combined with the predictable and reliable nature of the tidal resource, this flexible control can unlock economic opportunities [7,8].

The energy production potential of a tidal cycle, $E$, relates to the area $A$ of the enclosed basin and the tidal range $H$ via a relationship of the form $E \propto AH^2$ [9]. The eastern coast of the Irish Sea, UK, experiences some of the highest tidal ranges globally [2]. It is seen as a suitable location for the exploitation of this tidal range resource [10,11]. Nonetheless, the need to mitigate the typically high capital costs [12] whilst minimising the potentially significant environmental impact [13–15] has thus far hindered the commercial progress of UK-based proposals to date [1]. The development of a reliable UK renewable energy sector to meet carbon reduction targets [16] calls for research into the economically competitive implementation of tidal range power plants [12]. With an existing yet limited pool of schemes to draw knowledge from globally [1], numerical models have been developed to simulate design and operation, e.g., [6,17]. These provide a platform to investigate mechanisms in improving economic feasibility.

The current UK market trading framework for renewable energy schemes incentivises the maximisation of annual energy yield to increase economic gain [18]. A contract for difference (CfD) is agreed with the UK government, securing a fixed price of electricity in £/MWh, subsidising any income shortfall [19]. Most tidal power plant assessment studies assume a similarly incentivised economic model. Design optimisation typically centres on the characteristics and number of hydraulic structures (turbines, sluices, pumps), with the embankment path generally based on existing proposals and/or practical geographical limitations [10,20]. Meanwhile, the subsequent operational optimisation typically assumes an inflexible scheduling regime, where interval timings or water heads triggering operation mode transitions are fixed at all tidal cycles [21,22]. More recent studies reflect how the variability of the tidal elevation signal (e.g., its peaks and troughs) between adjacent tidal cycles incentivises a flexible operation in order to achieve maximum conversion of the energy resource. For example, gradient-based [6] and genetic algorithms [8] have been implemented to optimise operation scheduling of the Swansea Bay (and Cardiff [6]) proposal by Tidal Lagoon Power (TLP). Each successive tidal cycle was individually optimised to yield higher energy output than can be achieved through inflexible control scheduling. If we assume operation is driven by variable energy pricing markets that are sensitive to demand, then the potentially relatively small reductions in energy yield when deviating from the optimum scheduling for a given tidal cycle [23] suggests economic opportunities beyond maximising energy output. Harcourt et al. [7] investigated how exposing the TLP Swansea Bay lagoon proposal to wholesale energy markets would alter its optimal operation. The tidal lagoon operation was incentivised to generate during periods of high demand, and therefore higher sale price, in the day-ahead spot market. The resultant reduction in energy yield was offset by a higher income [7]. The potential for further opportunities afforded by flexible operation of tidal range schemes, in exploiting the volatility of wholesale energy markets, warrants analysis.

We seek to establish the practical feasibility of utilising tidal range energy in contributing a continuous power supply. Furthermore, we explore economic frameworks in which this would be a financially advantageous endeavour. The phasing out of fossil-fuel based energy sources in the UK requires a mix of both variable and dispatchable renewable generators [24]. Where the former (e.g., wind, solar, tidal stream) will likely provide the bulk of the required output, the latter (e.g., biomass, geothermal, stored energy) will be vital in providing resilience for a continuous supply of electricity [24,25]. The source of future baseload supply remains uncertain, given the projected decline in fossil fuel generators and potential restrictions on nuclear energy [26]. In addition, recent growth in UK variable renewable energy systems has increased grid congestion management costs, with further installation of unpredictable high penetration generators also potentially exacerbating balancing challenges [27]. An investigation into the degree to which tidal range power plants can generate continuously is therefore motivated by both reliably contributing to baseload power and improving grid integration. The application of tidal range energy for continuous generation has previously been explored through linked-based systems [28] and pumped storage [29], with each of these exhibiting certain bottlenecks. On the other hand, the intermittency of individual single-basin schemes could be offset by the conjunctive generation of a system of power plants that can exploit tidal phase differences.

Such a system could be supplemented by other technologies, e.g., tidal stream farms, in order to generate continuously [10,21]. To the best of the authors' knowledge, investigation of the flexible operation of a fleet of tidal range power plants has yet to be conducted in terms of their combined energy output and economic characteristics.

Characterising the interactions within a system of tidal range energy schemes is key to developing a numerical model that can in turn be used to cooperatively optimise operations. To date, modelling studies have simulated the operation of multiple schemes hydrodynamically with depth-averaged models [6,10,17,30]. Mackie et al. [31] considered a simplified reduction in electricity price when optimising the controls of a fleet of idealised tidal energy lagoons, requiring the system to be optimised cooperatively for maximum economic gain. In these cases, the application of multiple schemes was targeted at increasing the total energy output. In this study, we employ flexible operation to a fleet of tidal power plants so that they collectively pursue continuous generation when feasible. Three separate existing tidal power plant designs are considered, each proposed by a different company or authority. Nonetheless, in exploring economic opportunities, we consider that pursuing reliable continuous generation as a cooperative, non-competing operational body could increase their collective value, and thus concentrate on such a scenario.

## 2. Case Study

We investigate the potential for continuous generation in existing UK-based proposals. The tidal power plants selected are proposed for the north and south of Wales The four hour phase difference in tidal elevations between these two locations increases the potential cumulative generation window duration of a given day. This tidal phase difference provides an appropriate platform from which to employ flexible operation for continuous power generation. The position and layout of the selected design proposals are indicated in Figure 1:

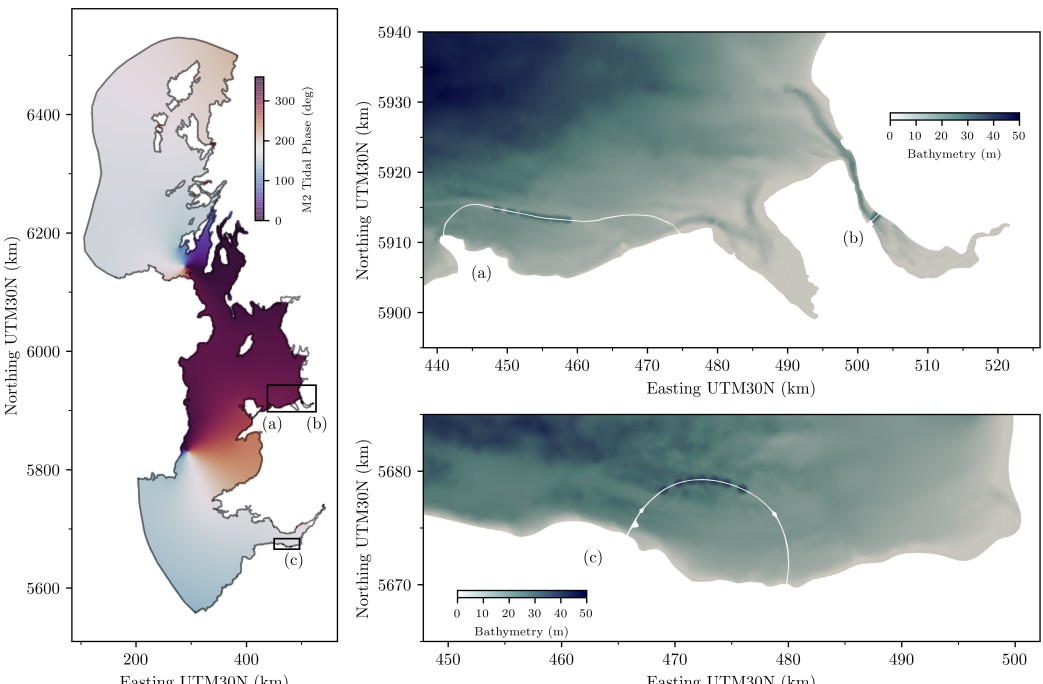

**Figure 1.** Location, $M_2$ tidal phase, bathymetry, design schematic and numerical model plan layout of three selected tidal range schemes: (**a**) Colwyn Bay Tidal Lagoon (CBL) [32,33], (**b**) Mersey Tidal Barrage (MB) [32,34] and (**c**) West Somerset Tidal Lagoon (WSL) [35,36].

(a)  A tidal lagoon in Colwyn Bay (CBL)—proposed by North Wales Tidal Energy [32,33].
(b)  The Mersey Tidal Barrage (MB)—where the Liverpool Combined Authority is exploring tidal range energy options [32,34].
(c)  The West Somerset Tidal Lagoon (WSL)—proposed by LongBay Seapower [35,36],

Design layout (Figure 1) and scheme characteristics (Table 1) are determined based on available information for the three tidal power plants. Embankment location for MB and WSL are derived directly from design schematics [32,35]. The ambiguous basin domain indicated for CBL [35] permits some design flexibility. Where information regarding hydraulic structure characteristics is unavailable, an assumption is made from similar power plant designs. For example, the turbine diameter for CBL is based on the diameter of the turbines of the TLP Swansea Bay lagoon design. Meanwhile, the number of sluices for MB and WSL are determined by a similar proportion between turbine and sluice numbers in CBL [37]. The basin area, tidal range and $M_2$ tidal phase $\phi$ (°) in Table 1 are calculated from outputs of the ambient flow conditions numerical model described in Section 3.1.

**Table 1.** Power plant specifications applied in this study.

| Power Plant: | (a) Colwyn Bay Tidal Lagoon | (b) Mersey Tidal Barrage | (c) West Somerset Tidal Lagoon |
|---|---|---|---|
| Notation | CBL | MB | WSL |
| Number turbines | 125 [37] | 28 [32] | 960 [32] |
| Number sluices | 40 [37] | 10 | 300 |
| Turbine diameter (m) | 7.35 | 8 [32] | 3.12 [32] |
| Turbine power output (MW) | 20 [32] | 25 [32] | 3.12 [32] |
| Installed capacity (MW) | 2500 | 700 | 2995.2 |
| Operation mode | Bi-directional | Bi-directional | Bi-directional |
| Basin area (km²) | 192.8 | 61.5 | 89.7 |
| Mean tidal range (m) | 6.77 | 7.65 | 8.27 |
| $M_2$ tidal phase (°) | 315.3 | 328.3 | 171.1 |

## 3. Methodology

The study applies numerical modelling to simulate the operation and hydrodynamic interaction of three tidal power plant proposals within the coastal ocean environment. Power plant controls are subsequently optimised and economic potential quantified by considering energy market data.

### 3.1. Hydrodynamic Model of the Irish Sea

Depth-averaged hydrodynamic modelling is applied in simulating tidal conditions around the power plant proposal locations. This method is selected to balance computational cost and accuracy. We employ the coastal flow solver *Thetis* (http://thetisproject.org/, [38,39]), implemented using the finite-element Partial Differential Equation (PDE) solver framework *Firedrake* (https://www.firedrakeproject.org/, [40]). The model is configured consistently with other depth-averaged studies applying *Thetis* to characterise the tidal dynamics for marine energy applications [6,7,13,28,41]. As such, it solves the non-conservative form of the shallow-water equations:

$$\frac{\partial \eta}{\partial t} + \nabla \cdot (H_d \mathbf{u}) \; = \; 0, \tag{1}$$

$$\frac{\partial \mathbf{u}}{\partial t} + \mathbf{u} \cdot \nabla \mathbf{u} - \nu \nabla^2 \mathbf{u} + f \mathbf{u}^\perp + g \nabla \eta \; = \; -\frac{\tau_b}{\rho H_d}, \tag{2}$$

$$\frac{\tau_b}{\rho} \; = \; g n^2 \frac{|\mathbf{u}|\mathbf{u}}{H_d^{\frac{1}{3}}}, \tag{3}$$

where $\eta$ represents free surface water elevation (m), $H_d$ total water depth (m), $\nu$ kinematic viscosity (m² s⁻¹) and $\mathbf{u}$ the depth-averaged velocity vector (ms⁻¹). Coriolis effects are characterised by $\mathbf{u}^\perp$, the velocity vector rotated counter-clockwise over 90°, and $f = 2\Omega\sin(\zeta)$, where $\Omega$ is the angular frequency of the Earth's rotation and $\zeta$ the latitude. Representation of bed shear stress, $\tau_b$ (kg m⁻¹ s⁻²),

employs the Manning coefficient $n$ (sm$^{-1/3}$). Wetting and drying processes are captured using a method formulated by Kärnä et al. [42]. A piecewise-linear discontinuous Galerkin finite element discretisation (DG-FEM) is configured through a $P_{1DG} - P_{1DG}$ velocity-free surface finite element pair. The solver adopts a semi-implicit Crank–Nicolson timestepping approach for temporal discretisation.

The domain spans the west coast of Great Britain and east coast of Ireland and is represented with an unstructured mesh (Figure 2), generated using *qmesh* (https://www.qmesh.org/, [43,44]). The bathymetry comprises three datasets. 20 m resolution LiDAR Composite DSM (Digital Surface Model) data from the Environment Agency [45] is used to map the coastlines where available. The remainder of the coastline and the ocean domain are covered by 1 Arc second and 6 Arc Second data obtained from the Edina Digimap Service [46]. All data are converted into a UTM zone 30N (EPSG:32630) projection. The model is forced at open ocean boundaries using eight tidal constituents from the TPXO database ($M_2, S_2, N_2, K_1, O_1, Q_1, P_1, K_2$) [4]. Finally, water volume fluxes at the Mersey and Severn rivers are imposed based on the National River Flow Archive [47].

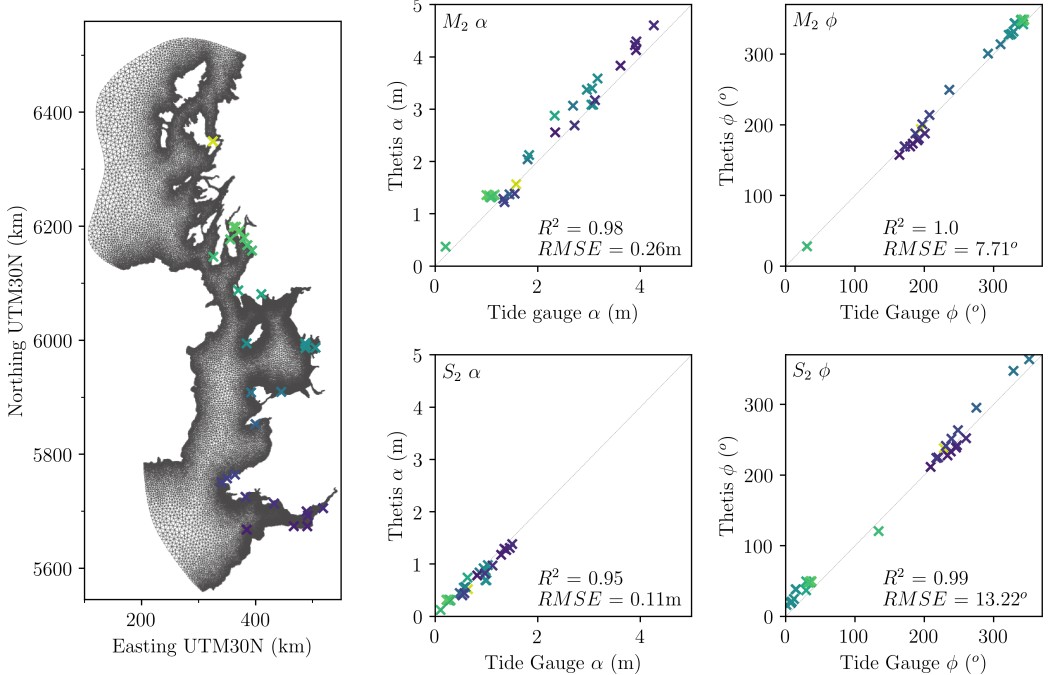

**Figure 2.** Unstructured mesh and computational domain including elevation data sampling locations for validation against BODC tide constituent data. $R^2$ correlation coefficient and Root Mean Square Error ($RMSE$) are computed between model and gauge data $M_2$ and $S_2$ tide constituent amplitude $\alpha$ (m) and phase $\phi$ (°).

The hydrodynamic model is validated against measured data prior to the inclusion of tidal power plants. A 30 day simulation of the ambient tidal conditions is conducted between 1 January 2018 and 30 January 2018, which represents sufficient time for the extraction of harmonic constituents from sampled temporal elevation data. In Figure 2, values of $M_2$ and $S_2$ amplitudes $\alpha$ (m) and phases $\phi$ (°) are extracted and directly compared against equivalent data provided by the British Oceanographic Data Centre (BODC) tide gauge network [48]. $R^2$ correlation coefficient and root mean squared error ($RMSE$) between measured and modelled data are indicated, their values suggesting sufficient performance of the hydrodynamic model in capturing the water elevations that drive the power plants.

### 3.2. Modelling Tidal Power Plants

0D modelling is employed to simulate tidal range scheme operation through its different "modes" and quantify energy output. Such modelling exercises are commonly applied in tidal range energy

assessment studies [5–7,10,21] due to their high computational efficiency. An equivalent 2D operation model would offer a limited benefit in terms of accuracy for the relatively small-scale schemes, whilst exhibiting a much higher computational cost [5]. The backwards finite difference model, which iteratively solves for the tidal power plant basin water elevation $\eta_i$ (m) at each timestep of $\Delta t = 100$ s, requires the following:

1. *Water level data:* Ambient tidal elevations $\eta_o$ (m) in each power plant location are extracted from outputs of the 2D Irish Sea model, harmonically reconstructed for a given duration at any point in time (Figure 3).
2. *Surface plan area data:* In applying the three tidal power plant basins to the computational mesh as subdomains, the manner in which the water surface plan $A$ (m$^2$) changes with basin free surface elevation $\eta_i$ (m) is determined for each tidal power plant (Figure 3). This provides a simplified means to represent intertidal regions.
3. *Hydraulic structure parameterisations:* Given operation mode and water elevations inside $\eta_i$ and outside $\eta_o$ the basin, head difference $H = \eta_i - \eta_o$ (m), flow rate $Q$ (m$^3$ s$^{-1}$) and power output $P$ (MW) through the turbines and sluices can be calculated where applicable. This allows energy yield $E$ (MWh) to be determined.
4. *Control schedule:* Based on either time intervals or specific head differences, the control schedule determines how the tidal power plant will shift between the different operation modes (holding, generating and sluicing in the ebb and flood phases) [5].

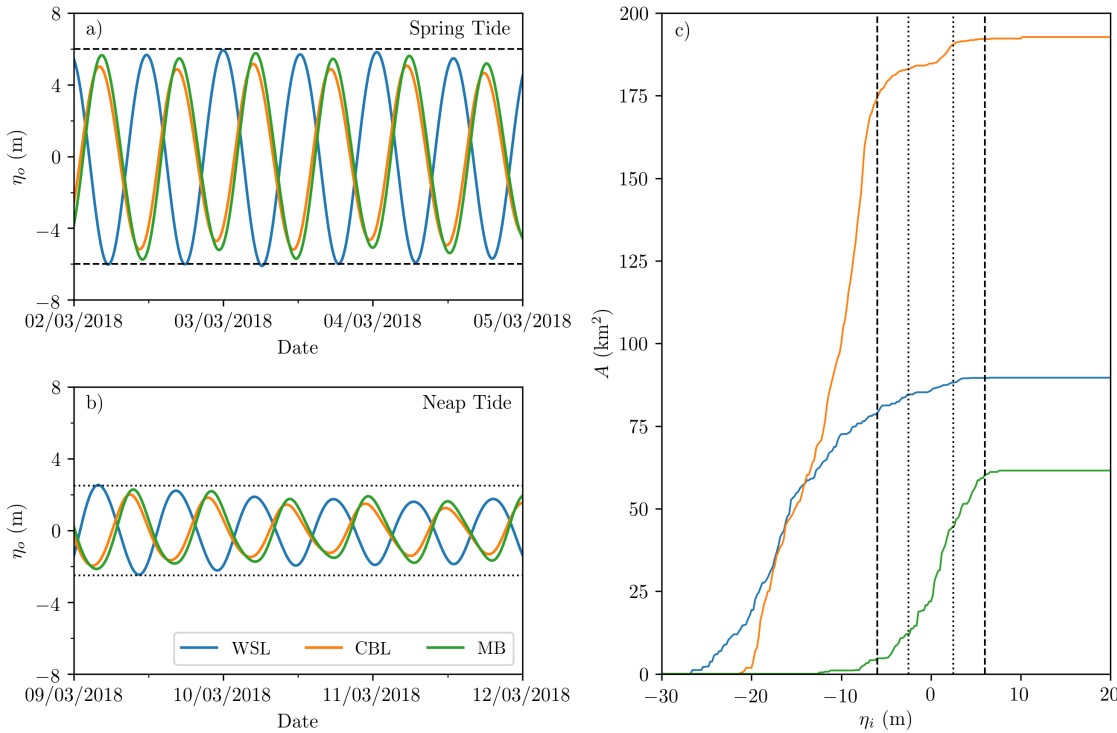

**Figure 3.** Ambient tidal elevations $\eta_o$ (m) in three scheme locations for (**a**) spring and (**b**) neap tides, and (**c**) the relationship between basin surface plan area $A$ (km$^2$) and basin elevation $\eta_i$ (m) in the corresponding tidal power plants.

A bi-directional, non-pumping operation regime is considered. Each tidal cycle consists of ebb/flood generation periods, simultaneous generation and sluicing (until turbine operation is no longer efficient), followed by sluicing until all hydraulic structures restrict water volume transfer. It is these latter, "holding", periods ($t_{h,e}$ and $t_{h,f}$) which are optimised in this study.

The generation-only period is set to four hours in the ebb and flood direction, with the other operation modes limited by practical constraints. Operation schedules exhibiting both flexibility and inflexibility between tidal cycles are determined, allowing or restricting changes in time, respectively. Initially, an inflexible operation regime is optimised for each power plant using a limited memory Broyden–Fletcher–Goldfarb–Shanno with limits (L-BFGS-B) algorithm [49]. Control periods $t_{h,e}$ and $t_{h,f}$ are stored in the vector $\boldsymbol{\tau}$ and applied to power plants individually in the following optimisation objective function:

$$\max_{\boldsymbol{\tau}} \quad \int_{t=0}^{t=t_s} P(\boldsymbol{\tau}, H, t)\, dt$$

$$\text{subject to} \quad \boldsymbol{\tau}_l \leq \boldsymbol{\tau} \leq \boldsymbol{\tau}_u$$

(4)

where $P$ is the total power output as calculated by the 0D model, while $\boldsymbol{\tau}_l$ and $\boldsymbol{\tau}_u$ represent the upper and lower limits of the decision variables, respectively. Each iteration simulates $t_s = 3$ months of power plant operation with the 0D model from 1 January 2018, a sufficient time period to capture local tidal elevations in determining inflexible control periods. The resultant optimised schedule returns the highest energy output predicted during the optimisation. This inflexible, energy maximised control schedule is herein denoted by EM-I with values indicated in Table 2.

**Table 2.** Summary of optimisation cases applied in this study. Unless classed as "Flexible", inflexible holding periods are presented in the following order: CBL $t_{h,e}$, CBL $t_{h,f}$, MB $t_{h,e}$, MB $t_{h,f}$, WSL $t_{h,e}$, WSL $t_{h,f}$.

| Optimisation Case | Objective Functional | Holding Periods (h) | Default $\delta$ | Cut-off Power $P_c$ (MW) |
|---|---|---|---|---|
| EM-I | Energy maximisation | 1.77, 1.65, 2.74, 2.14, 2.74, 2.19 | N/A | N/A |
| EM-F | Energy maximisation | Flexible | N/A | N/A |
| CG-I | Continuous generation | 1.00, 1.00, 3.00, 3.00, 3.00, 3.00 | N/A | N/A |
| CG-F$_{EM-I,0}$ | Continuous generation | Flexible | EM-I | 0 |
| CG-F$_{EM-I,75}$ | Continuous generation | Flexible | EM-I | 75 |
| CG-F$_{EM-I,150}$ | Continuous generation | Flexible | EM-I | 150 |
| CG-F$_{EM-I,225}$ | Continuous generation | Flexible | EM-I | 225 |
| CG-F$_{EM-I,300}$ | Continuous generation | Flexible | EM-I | 300 |
| CG-F$_{EM-F,0}$ | Continuous generation | Flexible | EM-F | 0 |
| CG-F$_{EM-F,75}$ | Continuous generation | Flexible | EM-F | 75 |
| CG-F$_{EM-F,150}$ | Continuous generation | Flexible | EM-F | 150 |
| CG-F$_{EM-F,225}$ | Continuous generation | Flexible | EM-F | 225 |
| CG-F$_{EM-F,300}$ | Continuous generation | Flexible | EM-F | 300 |
| CG-F$_{CG-I,0}$ | Continuous generation | Flexible | CG-I | 0 |
| CG-F$_{CG-I,75}$ | Continuous generation | Flexible | CG-I | 75 |
| CG-F$_{CG-I,150}$ | Continuous generation | Flexible | CG-I | 150 |
| CG-F$_{CG-I,225}$ | Continuous generation | Flexible | CG-I | 225 |
| CG-F$_{CG-I,300}$ | Continuous generation | Flexible | CG-I | 300 |

Determining the EM-I control sequence permits simulation of power plant operation with the 2D model. A domain decomposition technique is employed to model the structures, as described in Angeloudis et al. [17]. Calculations are conducted independently in the basin subdomains, and are coupled via the computation of fluxes through the hydraulic structures as per the 0D model [5,50]. The 2D model simulates hydrodynamics throughout the domain and the subdomains over a period of 30 days from 1 January 2018. Water elevation time series in the vicinity of the subdomains are then extracted for the harmonic analysis of the tidal signal. The latter is utilised for further 0D modelling and the optimisation in this study. Modelling the power plants in this way acknowledges the main hydrodynamic changes induced by their presence (i.e., the effects of these schemes on the tidal constituents).

*3.3. Control Schedule Optimisation*

Four different operational optimisation cases are investigated, building on the earlier study of Harcourt et al. [7]. Inflexible and flexible control of holding periods between tidal cycles is employed targeting both energy maximisation and continuous generation:

1.  *EM-I:* Energy maximisation, inflexible.
2.  *EM-F:* Energy maximisation, flexible.
3.  *CG-I:* Continuous generation, inflexible.
4.  *CG-F:* Continuous generation, flexible.

3.3.1. Energy Maximisation (EM)

The control schedule previously optimised for 2D modelling of the power plant system (Equation (4), Table 2) is considered as the inflexible energy maximisation case (EM-I). The following flexible optimisation functional maximising energy output (EM-F) is then applied individually to each power plant:

$$
\begin{aligned}
&\text{for } i \ = \ 1 : n_c \\
&\max_{\boldsymbol{\tau}_i} \quad \int_{t=i\times T}^{t=(i+1)\times T} P(\boldsymbol{\tau}_i, H, t)\, dt\ + \\
&\qquad\qquad \int_{t=(i+1)\times T}^{t=(i+2)\times T} P(\boldsymbol{\tau}_{EM\text{-}I}, H, t)\, dt \\
&\text{subject to} \quad \tau_l \leq \tau_i \leq \tau_u
\end{aligned}
\tag{5}
$$

In this method, control periods in sequential tidal cycles $\boldsymbol{\tau}_i$ are optimised iteratively for $n_c$ tidal cycles, each of duration $T \approx 12.42$ h. The functional simulates two cycles, the second imposing the inflexible energy maximisation (EM-I) holding periods, shown by $\boldsymbol{\tau}_{EM\text{-}I}$. Subsequently, once the flexible controls for the first cycle are established, the second one becomes the first in the next iteration, as the model steps forward in time. This approach, as introduced by Harcourt et al. [7], is adopted to disincentivise optimised values which may unbeknowingly be detrimental to the performance of the following cycle—or so-called "greedy generation". The initial guess for the optimised vector $\boldsymbol{\tau}_i$ is set equal to $\boldsymbol{\tau}_{EM\text{-}I}$.

3.3.2. Continuous Generation (CG)

Continuous generation is herein defined on a daily basis, where at any point during that period at least one power plant is generating. This reflects the manner in which the three tidal power plants are operating as a single cooperative system. Adopting this method highlights how cumulative continuous generation may not be possible for longer periods. Furthermore, a consistent rate of power output is not required in this study—a power output notably exceeding the baseload is anticipated given the variable nature of the resource. However, targeting continuous generation is expected to reduce large power output peaks [28].

An inflexible regime (CG-I) is established via trial and error, namely by imposing $t_{h,e} \ = \ t_{h,f} = 3$ h for MB and WSL and $t_{h,e} \ = \ t_{h,f} = 1$ h for CBL. In the case of the flexible continuous generation optimisation (CG-F) framework, the cumulative nature of continuous generation herein requires the three schemes to be optimised simultaneously. Therefore, $\boldsymbol{\tau}_i$ now contains six variables, i.e., $t_{h,e}$ and $t_{h,f}$ values for each of the three power plants (CBL, MB and WSL). A test is firstly conducted to establish

whether or not continuous generation is possible in a given tidal cycle. The following optimisation function to maximise generation time is imposed:

$$
\begin{aligned}
&\text{for } i = 1 : n_c \\
&\max_{\boldsymbol{\tau}_i} \qquad t_g(\boldsymbol{\tau}_i, H, t) dt, \\
&\text{subject to} \qquad \boldsymbol{\tau}_l \leq \boldsymbol{\tau}_i \leq \boldsymbol{\tau}_u
\end{aligned}
\tag{6}
$$

where $t_g$ is the total number of seconds when at least one of the power plants is generating during cycle $i$. The initial guess for each cycle is $\boldsymbol{\tau}_{CG\text{-}I}$ where CG-I is the inflexible continuous generation regime (CG-I).

If $t_g < T$, with $T$ = 44,700 s being the total number of seconds in a 12.42 h tidal cycle rounded to the nearest timestep ($\Delta t$ = 100 s), then the algorithm reverts to a default control vector. This default, denoted by $\delta$, can either be inflexible, pre-established values of $\boldsymbol{\tau}_i$ (EM-I and CG-I), or determined by applying the two-cycle energy maximisation function of Equation (5) (EM-F). An illustration of applying $\delta$ = EM-F, $\delta$ = EM-I and $\delta$ = CG-I is provided in Figure 4.

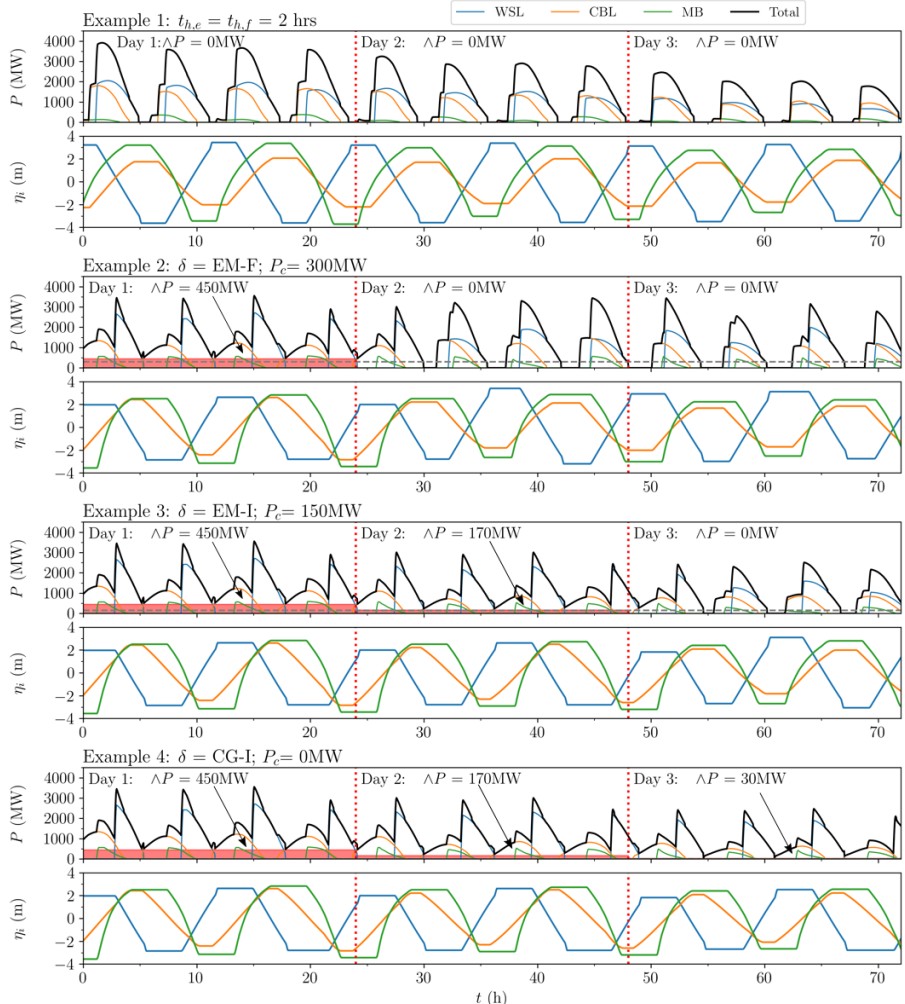

**Figure 4.** Power output $P$ (MW) and basin elevations $\eta_i$ (m) of the tidal power plants. Example 1 indicates a scenario where the holding periods $t_{h,e} = t_{h,f} = 2$ h for each scheme. Examples 2–4 display BG-F optimised schedules, applying different examples of "default" $\delta$, defined as the operation schedule reverted to when cut-off power $P_c$ (MW) cannot be achieved in the CG-F case. Daily minimum power $\wedge P$ (MW) determines the power sold on the Baseload Market (BM), indicated with red.

If $t_g = t_c$, then the control periods are reoptimised to maximise minimum cumulative power output $\wedge P$, once imposing the optimised control periods from Equation (6) as the initial guess in the functional:

$$\text{for } i = 1 : n_c$$

$$\max_{\tau_i} \int_{t=i \times T}^{t=(i+1) \times T} \wedge P(\tau_i, H, t) dt \tag{7}$$

$$\text{subject to} \quad \tau_l \leq \tau_i \leq \tau_u$$

A "cut-off" power, denoted by $P_c$ and illustrated in Figure 4, is employed, whereby the algorithm reverts to the aforementioned default $\delta$ if the minimum cumulative power output of the system does not reach this value (i.e., if $\wedge P < P_c$). If $\wedge P \geq P_c$, then the resultant $\tau_i$ forms the control periods for the tidal cycle. A "two-cycle" approach was considered, but found to offer no benefit to a control schedule incentivised to maximise $t_g$ and $\wedge P$ upon testing. The test in Equation (6) was found to be necessary, as an initial value of $\wedge P \geq 0$ is often required for Equation (7) to determine the optimum control periods.

### 3.4. Economics

Economic feasibility of the tidal power plant system is assessed by exposing the schemes to wholesale energy markets, thus linking energy output inherently to the volatility of price fluctuations. Income is therefore the primary indicator of the economic performance of the system. Given the known control schedules from the optimisation analyses (EM-I, EM-F, CG-I, CG-F), income can be calculated over the same period. The following pricing framework is devised to permit the investigation of scenarios where a greater spread of energy generation can be rewarded financially. Two types of electricity price time-series $p(t)$ (£/MWh) are considered, as displayed in Figure 5.

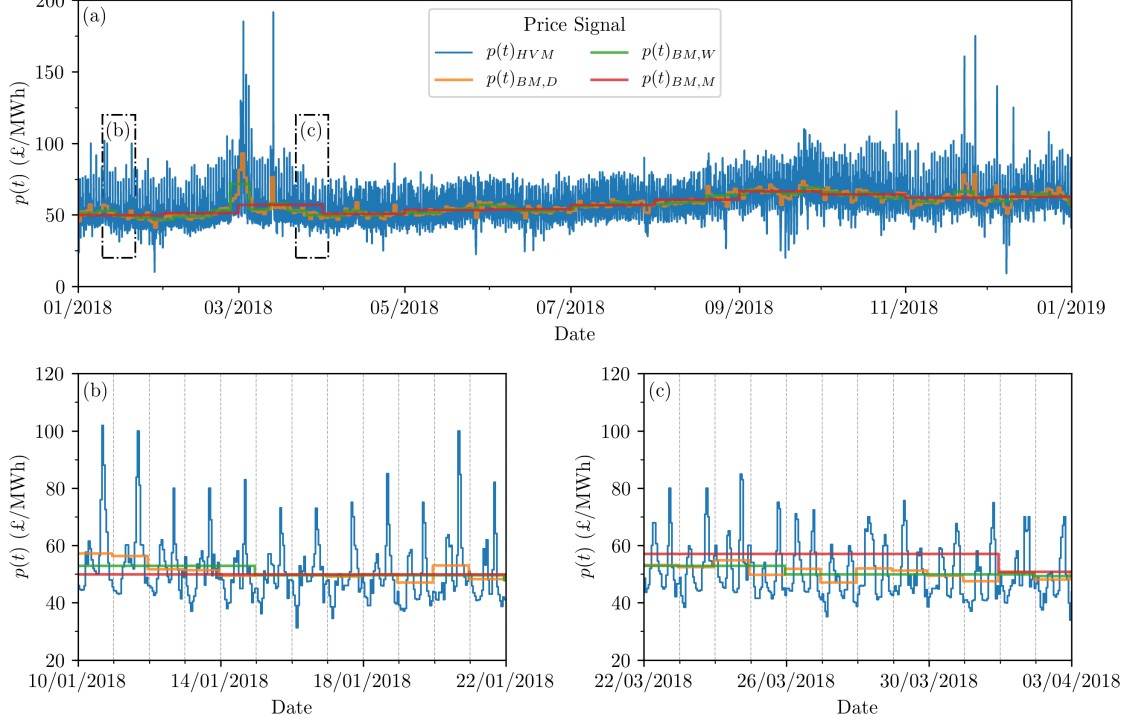

**Figure 5.** Hourly Variable $p(t)_{HVM}$ and corresponding Baseload Market price signals (£/MWh). $p(t)_{BM,D}$, $p(t)_{BM,W}$, $p(t)_{BM,M}$ refer to daily, weekly and monthly averaging respectively. The displayed snapshots indicate (**a**) the entirety of 2018, and periods where (**b**) $p(t)_{BM,D}$ and (**c**) $p(t)_{BM,M}$ would be financially advantageous.

1.  *Hourly Variable Market (HVM)*, $p(t)_{HVM}$ (£/MWh): This market is applied to reflect the value in providing power during periods of high demand. A price signal is derived from the Nord Pool N2EX market for the year 2018 [51]. The price of electricity changes hourly, with the tidal power system only needing to generate over the whole hour in question for valid market trading. While Nord Pool requires a constant rate for each hour, it is assumed that all energy generated by the power plant system in this hour is sold.

2.  *Baseload Market (BM)*, $p(t)_{BM}$ (£/MWh): This market considers a constant supply of electricity over the course of a day. Nord Pool does not regulate a UK-based baseload market [51]. Analysis of Ofgem baseload contract prices [52] reveals that monthly averages over a ten-year period are similar to equivalent monthly averages on the hourly Nord Pool N2EX market (HVM), albeit at a slight increase [51]. The employed BM data does therefore not directly reflect a historic price signal, but instead encompasses an averaging of the HVM price signal. Daily $p(t)_{BM,D}$, weekly $p(t)_{BM,W}$ and monthly $p(t)_{BM,M}$ averages are tested.

It is assumed, for simplicity, that the Hourly Variable Market price signal, and its processing for the Baseload Market, is forecast with 100% accuracy and without any alterations caused by the inclusion of the power plant system. Furthermore, trading on the BM is prioritised. If the tidal power plant system is able to deliver uninterrupted generation throughout an entire day, a constant energy output equivalent to the cumulative minimum power output $\wedge P$ of that day per timestep is traded on the BM. Figure 4 illustrates examples of this. Excess energy volume is traded on the HVM, including all the generated energy on a day where a minimum constant power output is not sustained, i.e., $\wedge P$ = 0 MW. This idealised energy market scenario provides a platform from which to explore the commercial viability of a power plant system targeting continuous generation.

*3.5. Optimisation Cases*

In investigating the combined economic potential for continuous power in the Colwyn Bay Tidal Lagoon (CBL), Mersey Tidal Barrage (MB) and West Somerset Tidal Lagoon (WSL), 18 control schedules are optimised to simulate operation with the 0D model between 1 January 2018 and 31 December 2018. A summary of these cases is provided in Table 2. For a baseline comparison commonly applied in academic literature and industry [5,6,8], inflexible (EM-I) and flexible (EM-F) energy maximisation cases apply the functions in Equations (4) and (5), respectively. Flexible continuous generation (CG-F) cases employ a combination of three "defaults" $\delta$ and five values of cut-off power $P_c$ (MW). For CG-F cases, the effect on operation and income by these two input parameters on Equations (6) and (7) is explored, denoted here by CG-F$_{\delta, P_c}$:

1.  *Default*, $\delta$: In the event of continuous generation not being permissible over a tidal cycle, the algorithm defaults to the system's secondary priority. Defaulting the cycle to maximise energy, as with $\delta$ = EM-I or $\delta$ = EM-F, considers a commitment to increasing the economic gain from that particular cycle. Meanwhile, $\delta$ = CG-I might prioritise phasing the system in a better position to attempt continuous generation in the next cycle.

2.  *Cut-off power*, $P_c$ *(MW)*: Employing a cut-off power $P_c$ recognises that, if only low levels of minimum cumulative power output $\wedge P$ are possible, it might not be worthwhile pursuing continuous generation. It also acknowledges that the required shift in $t_{h,e}$ and $t_{h,f}$ from the default might be large enough that the energy output is significantly reduced. $P_c$ values of 0 MW, 75 MW, 150 MW, 225 MW, and 300 MW are tested in the CG-F algorithm.

As such, a total of 15 CG-F control schedules are optimised, investigating (a) how the functionality of the optimisation algorithm, (b) operation characteristics of the tidal power plant system and (c) specialised trading in the energy markets that can be used to maximise economic gain through targeting continuous generation.

## 4. Results

### 4.1. Performance of System

The objective functions in Equations (4)–(7) do not include income as an input. Therefore, the power generation performance of the system can be analysed prior to implementing economic metrics. Here, we consider as annual (705 tidal cycles) performance indicators:

1. Total energy output, $E_T$ (TWh/year).
2. Number of "CG days" in which a continuous supply of energy is achieved.
3. Energy output on CG days to be traded on the Baseload Market, $E_{BM}$ (TWh/year), determined by the cumulative minimum power output $\wedge P$ of that day.
4. Average $\wedge P$ annually, $\overline{\wedge P}$ (MW).
5. Average $\wedge P$ on CG Days only, $\overline{\wedge P}_{CG}$ (MW).
6. Average power output peaks on all days, $\overline{\vee P}$ (MW).
7. Average power output peaks on CG days only, $\overline{\vee P}_{CG}$ (MW).

These performance indicators have been applied to present a diverse picture of power plant system generation characteristics under the different optimisation cases.

Table 3 and Figure 6 display results for the tidal power plant system under all cases. Comparing $E_T$ between the EM-I, EM-F, CG-I and CG-F cases provides an indication of the scale of the energy output compromise associated with incentivising the tidal power plant system to generate at a minimum cumulative power output of $\wedge P > 0$ MW each day. EM-F and EM-I, both not experiencing any CG days, yield a total of 19% and 10% more $E_T$ than the regime with the most CG Days, CG-F$_{CG-I,0}$, which achieves a degree of continuous generation on 242 days in the year. Furthermore, in all cases, the majority of energy is still traded on the hourly variable market as $E_{HV}$. $E_{BM}$ on CG-F$_{CG-I,0}$ is just 27% of $E_T$, despite 66% of the year generating at some level of continuous power. It can also be observed that the average daily peaks in power, both annually $\overline{\vee P}$ and exclusively on CG Days $\overline{\vee P}_{CG}$, are still significantly higher than the equivalent average daily minimum power outputs $\overline{\wedge P}$ and $\overline{\wedge P}_{CG}$. This indicates that, under CG-optimised control schedules, the cumulative power output is subject to a level of temporal variability such that peaks are much higher than $\wedge P$ for that day, and/or that low levels of $P$ are short-lived, but still determine the $\wedge P$ for that entire day. However, the difference between average peaks $\overline{\vee P}$ and average dips $\overline{\wedge P}$ over the year are notably reduced in CG cases compared to EM cases.

Figures 7–9 provide time-series illustrating how the individual power plants alter their holding periods $t_{h,e}$ and $t_{h,f}$ at each cycle when applying the flexible continuous generation (CG-F) optimisation functions. In addition, the transient tidal range is indicated at each site in addition to the daily rate of the minimum cumulative power output $\wedge P$. The indicated period is 1 January 2018 to 1 April 2018, with observed patterns present also for the remainder of the year. Certain observations are common to all three applied defaults ($\delta$) in CG-F. Days exhibiting $\wedge P > 0$ MW are achieved in clusters during spring tides. The peaks of these clusters are roughly proportional to the magnitude of the tidal range. This supports findings presented in Table 3, where a higher average maximum power output is achieved exclusively on CG Days $\overline{\vee P}_{CG}$ compared tp all days $\overline{\vee P}$, despite the former not being optimised to maximise energy output. This is due to the higher maximum power availability during spring tides, where the increased degree of flexibility also permits continuous generation. Furthermore, Figures 7–9 indicate the required control periods in generating continuously, where possible. For the Mersey Tidal Barrage (MB) and West Somerset Tidal Lagoon (WSL), $t_{h,e}$ and $t_{h,f}$ values in $\tau$ are pushed towards the upper limit of $\tau_u = 4$ h (flood periods $t_{h,f}$ in particular for WSL), while the Colwyn Bay Tidal Lagoon (CBL) gradates (ebb periods $t_{h,f}$ in particular) towards the lower limit $\tau_l = 0$ h.

**Table 3.** Energy and power output results of all optimisation cases over 705 tidal cycles ($\approx$1 year). CG-F subscript indicates associated $\delta$ and cut-off power $P_c$ (MW). Days where continuous generation is achieved are indicated as CG Days, as well as the associated proportion of the total energy $E_T$ (TWh/year) to be traded in the idealised baseload market $E_{BM}$ (TWh/year). $\overline{\wedge P}$ and $\overline{\wedge P}_{CG}$ (MW) indicate average daily minimum cumulative power output over the year and exclusively on CG Days, respectively, while $\overline{\vee P}$ and $\overline{\vee P}_{CG}$ (MW) show equivalent daily average power peaks. The combined installed capacity of the power plant system is 6195.2 MW.

| Optimisation Function | CG Days | $E_{BM}$ (TWh/year) | $E_T$ (TWh/year) | $\overline{\wedge P}$ MW | $\overline{\wedge P}_{CG}$ MW | $\overline{\vee P}$ MW | $\overline{\vee P}_{CG}$ MW | $\overline{\vee P} - \overline{\wedge P}$ MW | $\overline{\vee P}_{CG} - \overline{\wedge P}_{CG}$ MW |
|---|---|---|---|---|---|---|---|---|---|
| EM-I | 0 | 0.00 | 12.18 | 0 | N/A | 3439.6 | N/A | 3439.6 | N/A |
| EM-F | 0 | 0.00 | 13.20 | 0 | N/A | 3528.8 | N/A | 3528.8 | N/A |
| CG-I | 200 | 2.08 | 11.39 | 237.3 | 433.1 | 3179.8 | 3891.0 | 2942.5 | 3457.9 |
| CG-F$_{EM\text{-}I,0}$ | 218 | 2.76 | 11.05 | 315.5 | 528.2 | 3040.5 | 3630.3 | 2725.0 | 3102.1 |
| CG-F$_{EM\text{-}I,75}$ | 209 | 2.73 | 11.06 | 312.1 | 545.1 | 3040.8 | 3669.0 | 2728.7 | 3123.8 |
| CG-F$_{EM\text{-}I,150}$ | 194 | 2.67 | 11.09 | 304.3 | 572.3 | 3044.6 | 3729.5 | 2740.3 | 3157.0 |
| CG-F$_{EM\text{-}I,225}$ | 174 | 2.50 | 11.17 | 285.6 | 599.1 | 3063.4 | 3793.8 | 2777.8 | 3194.7 |
| CG-F$_{EM\text{-}I,300}$ | 140 | 2.19 | 11.31 | 250.0 | 651.9 | 3111.2 | 3891.5 | 2861.2 | 3239.6 |
| CG-F$_{EM\text{-}F,0}$ | 206 | 2.63 | 11.55 | 299.8 | 531.1 | 3286.7 | 3645.6 | 2986.9 | 3114.5 |
| CG-F$_{EM\text{-}F,75}$ | 173 | 2.22 | 11.80 | 253.9 | 535.6 | 3331.2 | 3658.9 | 3077.3 | 3123.3 |
| CG-F$_{EM\text{-}F,150}$ | 102 | 1.27 | 12.36 | 145.3 | 520.0 | 3413.9 | 3650.4 | 3268.6 | 3130.4 |
| CG-F$_{EM\text{-}F,225}$ | 18 | 0.26 | 12.98 | 29.1 | 590.5 | 3515.4 | 3752.6 | 3486.3 | 3162.0 |
| CG-F$_{EM\text{-}F,300}$ | 6 | 0.11 | 13.09 | 12.0 | 730.8 | 3529.3 | 3902.8 | 3517.3 | 3172.0 |
| CG-F$_{CG\text{-}I,0}$ | 242 | 2.94 | 11.08 | 335.8 | 506.5 | 3071.5 | 3600.7 | 2735.7 | 3094.1 |
| CG-F$_{CG\text{-}I,75}$ | 233 | 2.92 | 11.09 | 333.5 | 522.4 | 3074.1 | 3636.5 | 2740.6 | 3114.1 |
| CG-F$_{CG\text{-}I,150}$ | 224 | 2.89 | 11.10 | 330.0 | 537.7 | 3077.2 | 3676.3 | 2747.2 | 3138.6 |
| CG-F$_{CG\text{-}I,225}$ | 209 | 2.80 | 11.13 | 320.0 | 558.8 | 3082.6 | 3716.9 | 2762.6 | 3158.1 |
| CG-F$_{CG\text{-}I,300}$ | 203 | 2.72 | 11.15 | 310.0 | 557.4 | 3090.2 | 3731.7 | 2780.3 | 3174.4 |

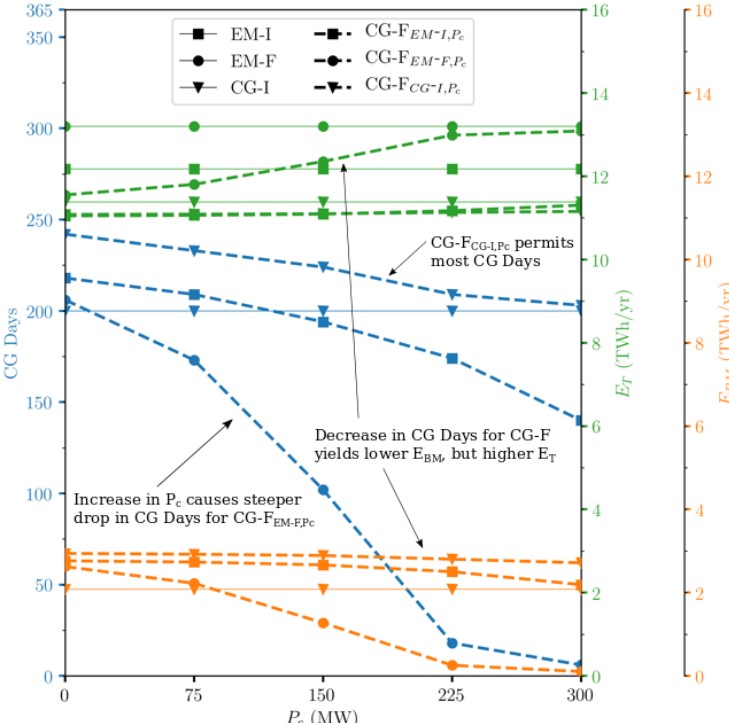

**Figure 6.** Annual (705 tidal cycles) energy results of optimisation cases from Table 3: total energy output $E_T$ (TWh/year), total Baseload Market output $E_{BM}$ (TWh/year) and number of days where $\wedge P > 0$ MW.

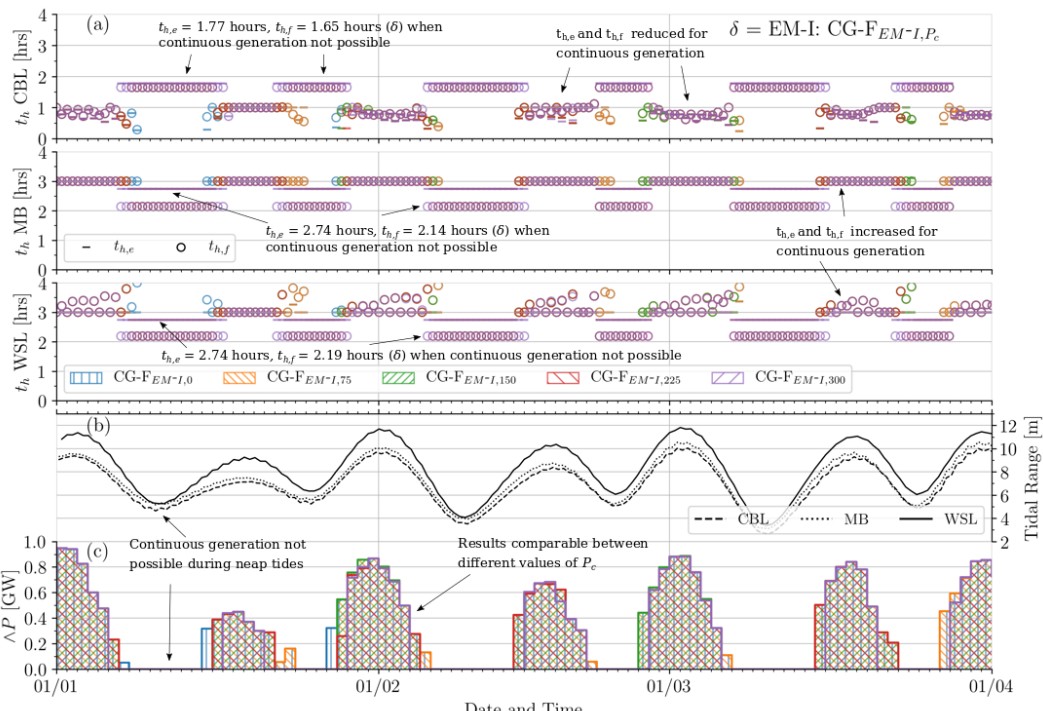

**Figure 7.** Temporal operation characteristics of the tidal power plant system in applying flexible controls to optimise towards continuous generation, with $\delta$ = EM-I, an inflexible energy maximisation default (CG-I$_{EM-I,P_c}$). For all three power plants, (**a**) indicates holding periods $t_{h,e}$ and $t_{h,f}$ per tidal cycle (**b**) the tidal range each cycle and (**c**) daily cumulative minimum power output $\wedge P$ (GW).

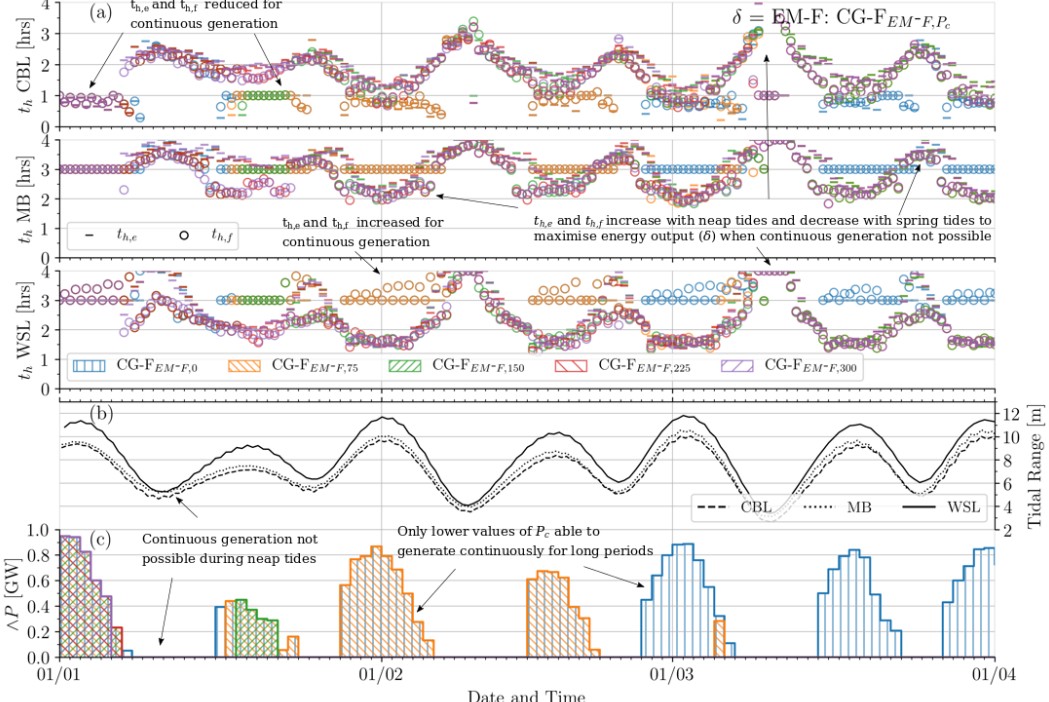

**Figure 8.** Temporal operation characteristics of the tidal power plant system in applying flexible controls to optimise towards continuous generation, with $\delta$ = EM-F, a flexible energy maximisation default (CG-I$_{EM-F,P_c}$). For all three power plants, (**a**) indicates holding periods $t_{h,e}$ and $t_{h,f}$ per tidal cycle (**b**) the tidal range each cycle and (**c**) daily cumulative minimum power output $\wedge P$ (GW).

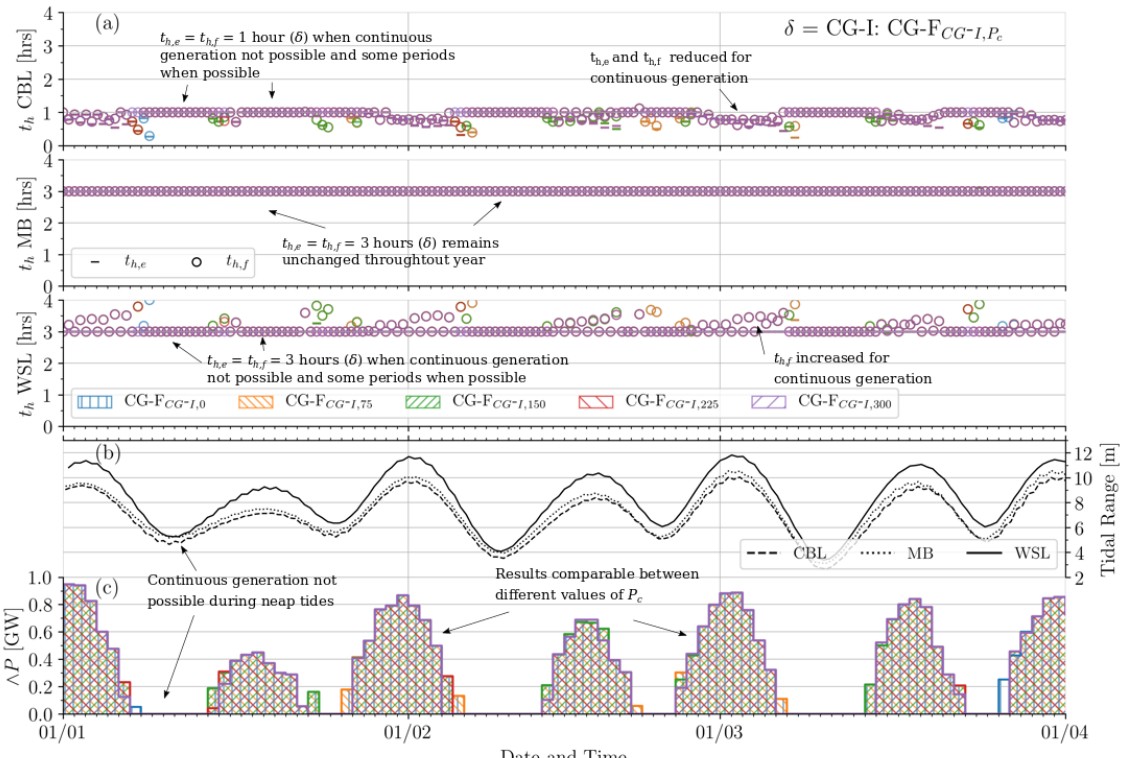

**Figure 9.** Temporal operation characteristics of the tidal power plant system in applying flexible controls to optimise towards continuous generation, with $\delta$ = CG-I, an inflexible baseload generation default (CG-I$_{CG-I,P_c}$). For all three power plants, (**a**) indicates holding periods $t_{h,e}$ and $t_{h,f}$ per tidal cycle (**b**) the tidal range each cycle and (**c**) daily cumulative minimum power output $\wedge P$ (GW).

The manner in which Figures 7–9 differ provides further insight into how the default $\delta$ and cut-off power $P_c$ affect operation of the power plant system when optimising flexible control periods to target continuous generation (CG-F). The default $\delta$ is applied to acknowledge that the operation in a given tidal cycle affects the performance in the subsequent one. This can potentially eliminate the ability to achieve $\wedge P > P_c$ in the latter. Applying $\delta$ = CG-I (Figure 9) permits all power plants to generate continuously between sequential cycles to a marginally greater extent than $\delta$ = EM-I (Figure 7). Results in these two cases are similar at different values of $P_c$. Meanwhile, when applying $\delta$ = EM-F (Figure 8), holding periods in cycles where continuous generation is not possible (neap tides) increase their duration to maximise energy. In turn, this restricts the ability to continuously generate ($\wedge P > 0$ MW) in subsequent cycles. This is largely due the disparity between CBL holding periods where energy maximisation is defaulted to, and the required holding periods for continuous generation. The effect described is perpetuated at higher values of $P_c$, leading to a chain of missed opportunities to continuously generate. It can be seen in Figure 8 that whole blocks of $\wedge P$ are not possible at higher values of $P_c$, despite $\wedge P$ in these time periods being much higher than $P_c$ in equivalent blocks for $\delta$ = EM-I and $\delta$ = CG-I. A much higher tidal range is required to then re-establish a chain of days where $\wedge P$ is achieved. This emphasises the significance of operation controls in preceding cycles, whereby long periods of potentially continuous generation may not be achieved.

### 4.2. Income

The economic performance between optimisation cases can be compared following control schedule optimisation. Section 3.4 provides an overview of the energy market framework applied in this study, including the three idealised Baseload Market (BM) models designed upon daily, weekly and monthly averaging of the Hourly Variable Market (HVM). As previously noted, a penalty in energy output is consistently experienced when seeking to maximise the minimum cumulative

power output $\wedge P$ on a daily basis. We therefore explore the manner in which energy market trading can be applied to economically incentivise optimising towards more continuous, albeit less energetic, operation regimes.

Figure 10 presents annual tidal power plant system Baseload Market (BM) income $I_{BM}$ and total income $I_T$ of all optimisation control schedules. The central *y*-axis on each figure indicates application of the unaltered, averaged pricing structures outlined in Section 3.4. CG-I and CG-F optimisation cases consistently produce less income $I_T$ than the EM-I and EM-F cases in the power plant system, implying that under unchanged financial conditions they are less financially lucrative. A scaling $\alpha$ of the BM pricing $p(t)_{BM}$ is explored. In Figure 10, we determine the value of $\alpha$ required to make CG-F optimisation a more re-numerative endeavour than the EM cases. It can be seen that CG-F cases produce a higher total income $I_t$ than both EM cases between $\alpha \approx 1.75$ to $1.9$ for CG-F$_{EM-I,P_c}$ cases, $\alpha \approx 1.65$ for CG-F$_{EM-F,P_c}$ cases and between $\alpha \approx 1.7$ to $1.75$ for CG-F$_{CG-I,P_c}$ cases. Under the applied idealised market framework, a notable amplification of Baseload Market price $p(t)_{BM}$is thus required to financially reward a more continuous operation regime.

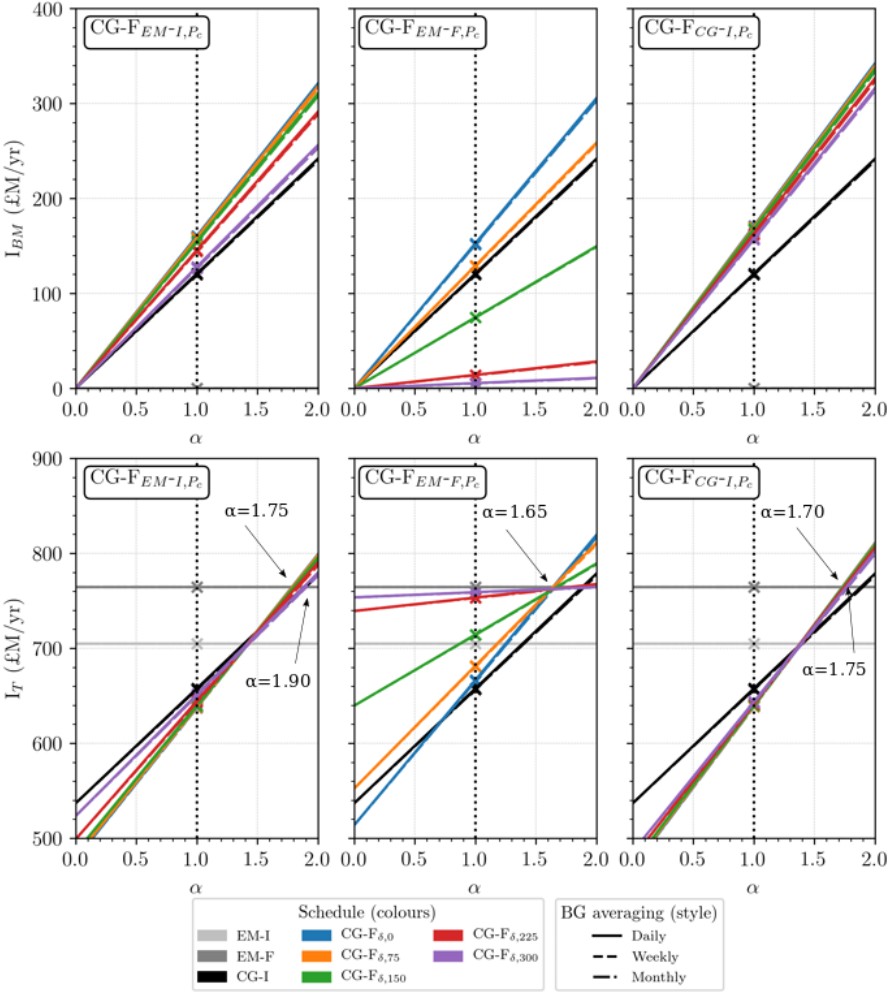

**Figure 10.** How scaling $\alpha$ of the Baseload Market electricty price $p(t)_{BM}$ (£/MWh) affects income both exclusively in the Baseload Market (BM), and in combination with the Hourly Variable Market (HVM). l-r is $\delta$ = EM-I, $\delta$ = EM-F, and $\delta$ = BG-I default cases.

Figure 11 indicates the minimal difference in annual Baseload Market (BM) income $I_{BM}$ and total income $I_T$ between daily $p_{BM,D}$, weekly $p_{BM,W}$ and monthly $p_{BM,M}$ BM price signals. However, understanding the provenance of these differences could be useful in predicting the effect

on $I_{BM}$ and $I_T$ in processing future Hourly Variable Market (HVM) price signal exhibiting a different structure. Figure 11 shows daily sums of $I_{BM}$ and $I_T$ when applying the corresponding averaged cases. It can be seen that, on days where $\wedge P > 0$ MW (i.e., $I_{BM} > 0$ £/MWh), there is little disparity in $I_{BM}$ and $I_T$ among averaging cases. The percentage difference between the daily baseload average case and the weekly and monthly averages is consistently less than 10% for $I_{BM}$, and often less than 5% in $I_T$. The weekly and monthly averages are less susceptible to dips in hourly variable electricity price, but also less able to take advantage of large spikes in price, as experienced in March 2018. Indeed, while a sudden spike may be advantageous to the day in question on the daily averaging market, its price benefits on the weekly and monthly markets are more spread out. In the event of baseload market scaling (the application of $\alpha > 1$), these effects would be amplified.

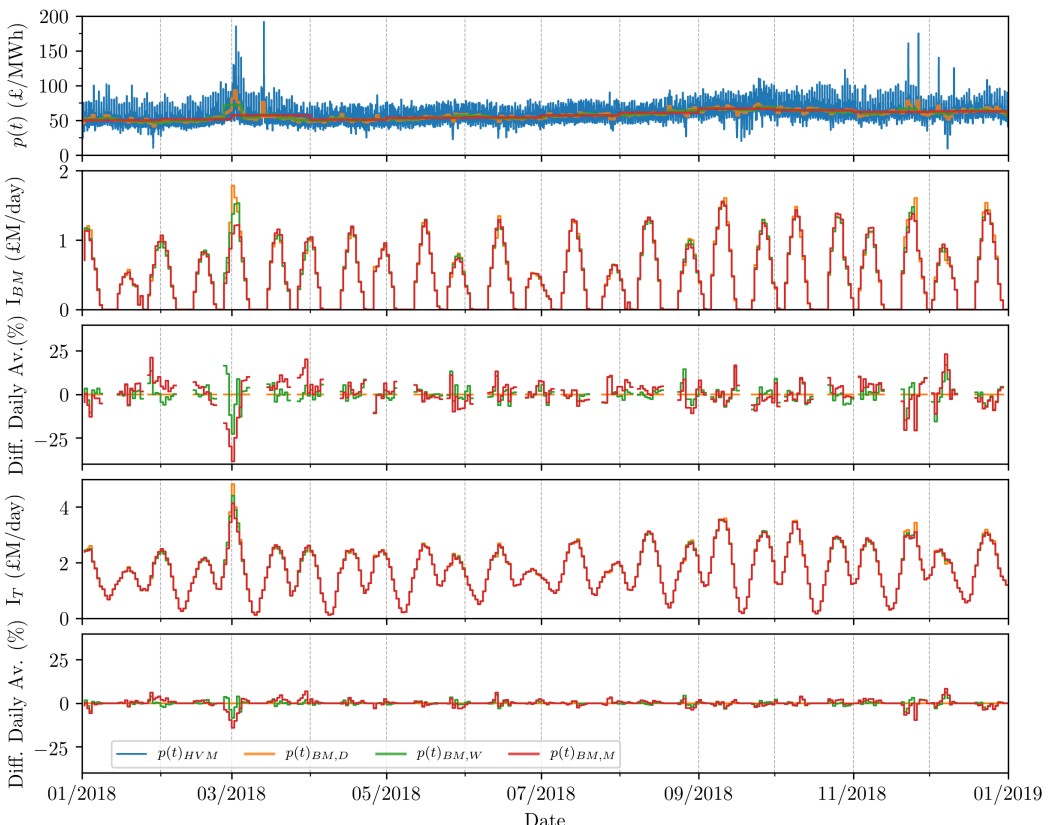

**Figure 11.** Tidal range power plant system daily income upon trading in baseload markets $I_{BM}$ and hourly variable markets $I_{HVM}$ ($I_{BM} + I_{HVM} = I_T$). Displayed results are from CG-F$_{CG-I,0}$ control schedule, which yields the highest baseload energy $E_{CG}$ of all optimisation cases. Percentage difference between daily averaged baseload price and the weekly and monthly equivalents is provided.

## 5. Discussion

### 5.1. Continuous Generation Potential

The following discussion focuses on the degree to which the modelled tidal range power plant system can generate continuously, and the resulting implications on its value in the UK energy market. Initially, we assess the application of tidal range energy as a reliable alternative to commonly applied generators which provide baseload supply in the UK, such as nuclear power. Considering uncertainties in future demand and potential expansion restrictions and/or diminishing availability of nuclear power stations, the longevity of tidal range constructions adds further value in their potential contributions [26]. Targeting continuous extraction of the semi-diurnal tidal resource lends itself to being applied and assessed on a daily basis (two whole tidal cycles). A number of optimisation cases

geared towards generating at a cumulative minimum power output of $\wedge P > P_c$ are tested. In turn, a number of tidal power plant performance indicators are derived from the results (Table 3). One such indicator is the number of days when a degree of continuous generation is achieved. Results indicate a maximum of 242 days in the year considered. Therefore, the current 6195.3 MW capacity setup demonstrates an ability to reliably make frequent contributions to baseload supply at a daily average of $\overline{\wedge P}_{BG} > 500$ MW. However, this is is insufficient as a consistent contributor to baseload power. This is further supported by the $\wedge P$ times series in Figures 7–9, where $\wedge P = 0$ MW occurs cyclically during neap tides. While expanding the system across a wider tidal phase window could prompt further system flexibility, it is demonstrated in Figure 1 that the available phase difference in the Irish Sea does not exceed the four hours exploited with the schemes in this case study. The additional implementation of tidal range energy developments in other common power plant proposal locations, such as the Solway Firth or other parts of the Severn Estuary [12], would therefore offer limited further phase diversity. However, Example 1 in Figure 4 illustrates how, under a consistent operation regime of $t_{h,e} = t_{h,f} = 2$ h for all power plants, the commencement and conclusion of generation between schemes at each cycle has a smaller total phase difference ($\approx 2$ h) than is observed in Figure 3a,b ($\approx 4$ h). It can also be observed in Figure 2 how there exists a similar phase difference north and south of Wales in both the $M_2$ and $S_2$ tidal constituents. Therefore, characteristics of individual power plants such as bathymetry and the number and size of hydraulic structures could be affecting operation. A sensitivity study of the current setup with regards to these parameters and/or testing of alternative idealised or existing proposals could investigate this further.

Operational analysis of the applied tidal power plant system provides information about how the combined setup could offer an increased duration of baseload supply. It can be observed in Figures 7–9 that, when the tidal range in all power plant locations $\lesssim$ 6m (i.e., neap tides), continuous generation is not possible in any optimisation cases. The implementation of pumping, to reduce low tide and increase high tide in the enclosed basin [53], could increase the achievable head difference, potentially negating the limitations of the resource. Furthermore, combining the system with energy storage and/or more commonly used dispatchable generators such as biomass would ensure a continuous supply. A similar end could be attained by deploying strategically located tidal stream arrays to fill in the gaps of cumulative intermittency of the power plant system.

Controlling the tidal power plant system in the partially continuous manner demonstrated in this study presents opportunities beyond contributing to baseload supply, owing to resource and operation predictability. Firstly, the daily variability in power output is reduced when pursuing continuous generation. Table 3 illustrates multiple performance indicators, including the disparity between average daily maximum and minimum power on CG Days ($\overline{\vee P}_{CG} - \overline{\wedge P}_{CG}$). Results indicate that, while there still lies a significant power variability on these days, the effect on the whole year ($\overline{\vee P} - \overline{\wedge P}$) in comparison to EM cases is notable. Secondly, exploring the impact of a default $\delta$—the optimisation function or pre-determined control sequence when continuous generation is not possible—highlights the flexibility of the system. By applying a default in which schemes are optimised to maximise energy, we demonstrate how the system can be used for different purposes in sequential cycles (Figures 7 and 8). As noted previously, the effect on the continuous generation potential of subsequent cycles must be considered. The application of a cut-off power $P_c$, the minimum allowable cumulative power output before reverting to this default, is shown to dismiss continuous generation on less energetic tidal cycles where efforts could be better placed towards energy maximisation. This could be a useful metric for the flexible system to assess the necessity in generating continuously on a given day. These two points illustrate how modelled tidal range system provides a service that lies between outright baseload supply and maximising energy to meet specific demand patterns. Such a system could be an advantageous asset to the UK. A report into hydropower by Harby et al. [54] for IEA Hydropower outlines the manner in which further installation of UK-based variable renewable energy generators will require additional investments in flexible infrastructure. In constructing and operating

the modelled tidal range system, the integration challenges triggered by the power volatility of other generators could be curtailed through the degree of flexibility afforded by the system.

### 5.2. Economics

Cumulative income is applied as the sole metric of economic performance for the tidal power plant system. A reduction in energy output is naturally experienced when optimising operation schedules to target continuous generation (instead of maximising energy). We therefore assess exposing the power plant system to wholesale energy markets based on realistic, demand-driven price data in £/MWh. The applied economic framework splits energy market trading between baseload (Baseload Market: BM) and excess supply (Hourly Variable Market: HVM). The BM varies either daily, weekly or monthly and the HVM varies hourly (Figures 5 and 11). Optimised schedules which model the power plant system trading solely in the HVM (as with both energy maximisation cases EM-I and EM-F, Table 3), demonstrate a scenario where schemes sell energy independently. Where a cumulative continuous generation is achieved on at least one day in the year, the individual schemes are assumed to trade energy as a singular body. By applying BM price scaling $\alpha$, we assess the point at which this becomes profitable, as illustrated in Figure 10. At scaling values between $\alpha = 1.65$ and $1.90$, this represents a fairly significant increase in required electricity price. In establishing the source of this income shortfall, we speculate potential alignment with the contract for difference model (CfD), which is commonly applied to variable renewable generators. Such renumeration could therefore be in the interest of the UK government, in a similar manner to CfDs subsidising renewable generators.

Considering factors beyond direct income could highlight further economic advantages in optimising operation to target continuous generation. The indispatchability of high penetration variable renewable generators such as wind and solar energy can amplify short-term grid integration costs, which finance congestion management and balancing [27]. A predictable tidal range energy system incentivised to spread its naturally intermittent generation could reduce these associated costs, in turn curbing electricity price volatility once a comparable capacity is installed [55]. In addition, the requirement of storage facilities (e.g., batteries) is reduced. Future investigations could implement these financial considerations, assessing the reduction in a required subsidy. Additionally, a quantification of battery sizes for complementing the variable power peaks of the tidal energy system could further improve economic feasibility.

### 5.3. Optimisation Methodology

The optimised framework applies a number of variable inputs ($\delta$, $P_c$) in assessing how it is best set up to provide continuous power practically and economically. We discuss here potential amendments to improve its functionality to further assess the feasibility in continuous generation. The use of pumping to increase the duration of continuous power periods in the year is proposed earlier in the discussion. Adding pumping periods in the decision variables of $\tau$ would likely be required given the variable tidal signal constantly changing the needs of the schedule. This would significantly increase the computational time. A method to curb this extra computational cost could be to reduce the number of power plants modelled. Mersey Tidal Barrage could be eliminated from the system, as it harnesses a similarly phased tidal resource to the higher capacity Colwyn Bay Tidal Lagoon. Simulating the operation of just two power plants could also allow testing of optimising two cycles (one day) at a time. This would remove the possibility of a "morning" cycle being scheduled to generate at a minimum power output $\wedge P > 0$ MW, but then avoid trading on the Baseload Market as the subsequent "evening" cycle on the same day is unable to do so. Optimising on a day-by-day basis would double the number of decision variables of $\tau$, and therefore only be computationally efficient when determining solely holding periods, excluding pumping. Applying the maximisation of income as the objective function, given certain Baseload Market conditions, could also be an option, as well as integrating expenditure through capital, maintenance and operation costs. Idealised, randomised or realistic

supply and demand requirements, as opposed to consistent energy maximisation or continuous generation incentives, would further assess the flexibility of the system.

## 6. Conclusions

This study explores the potential to operate three existing tidal range power plant proposals in the Irish Sea as a single, flexible system geared towards providing continuous generation. Operational and economic performance indicators are compared between traditional energy maximisation (EM) control schedules and operation regimes optimised to generate continuously (CG). For both EM and CG optimisation cases, the ability to exert flexible or inflexible control of tidal power plant holding periods $t_{h,e}$ and $t_{h,f}$ between tidal cycles is applied. The following section outlines key conclusions that can be derived from the results and subsequent discussion.

The ability of the system to generate continuously is assessed daily, depending on whether minimum cumulative continuous power output $\wedge P > 0$ MW. Under the applied conditions, neither of the EM cases generates continuously for any days in the year. Continuous generation is attainable to a degree in all CG optimisation cases, for a maximum of 66% of the days in the year considered. During these "CG days", the average daily minimum cumulative power output $\overline{\wedge P}_{CG} > 500$ MW (out of 6195.3 MW total installed capacity) in all 15 CG cases with flexible control periods (CG-F). Optimising to target continuous generation also notably reduces the difference between average daily minimum and maximum cumulative power output over the entire year $\overline{\vee P} - \overline{\wedge P}$ compared to energy maximisation cases. The operational flexibility afforded by individual tidal range schemes plays a key role in the ability of the cumulative system to collectively generate continuously. In a given tidal cycle, a higher tidal range offers more flexibility, with tidal cycles where it is not possible for continuous generation to occur during neap tides. The power plant system operation during cycles where continuous generation is not achievable (i.e., the default $\delta$) greatly affects the ability of the subsequent cycle to generate continuously. For these non-continuous cycles, employing holding periods in the vicinity of values typically used when generating continuously permits a higher number of "CG days", and overall generates a higher volume of daily continuous energy $E_{CG}$.

An idealised energy market framework is employed to economically assess the optimised control schedules. Two energy markets are considered, reflecting baseload and variable excess power output of the system. It is found that scaling of the baseload market would be required to make the combined power plant system competitive compared to equivalent schemes operating individually to maximise energy. Such scaling could practically be applied as subsidies, which would be in the interest of operators in ensuring reliability, reducing grid integration management costs and potentially reducing energy market price fluctuations. Further economic considerations based on reduced balancing and storage costs of the flexible system must be acknowledged. Options to improve annual generation duration are suggested, including implementing pumping intervals and supplementing the system with alternative generators. It is argued that the value of operating the system in this way lies in reliably contributing to baseload supply, reducing the difference between power output peaks and dips, and delivering flexible demand response.

**Author Contributions:** Conceptualization, L.M. and D.C. and M.P. and A.A.; methodology, L.M. and A.A.; software, L.M.; validation, L.M.; formal analysis, L.M.; investigation, L.M.; resources, L.M.; data curation, L.M.; writing—original draft preparation, L.M.; writing—review and editing, L.M. and D.C. and M.P. and A.A.; visualization, L.M.; supervision, M.P. and A.A.; project administration, M.P. and A.A.; funding acquisition, M.P. and A.A. All authors have read and agreed to the published version of the manuscript.

**Funding:** L.M. would like to acknowledge the financial support of an EPSRC studentship award. D.C. acknowledges the financial support of the Tidal Stream Industry Energiser project (TIGER), co-financed by the European Regional Development Fund through the Interreg France (Channel) England Programme. A.A. acknowledges the support of NERC through the Industrial Innovation fellowship grant NE/R013209/2. M.D.P. acknowledges support from EPSRC under grants EP/M011054/1, EP/L000407/1, EP/R029423/1.

**Conflicts of Interest:** The authors declare no conflict of interest. The funders had no role in the design of the study; in the collection, analyses, or interpretation of data; in the writing of the manuscript, or in the decision to publish the results.

## Abbreviations

The following abbreviations are used in this manuscript:

TLP        Tidal Lagoon Power Ltd.
CfD        Contract for Difference
CBL        Colwyn Bay Tidal Lagoon
MB        Mersey Tidal Barrage
WSL        West Somerset Tidal Lagoon
EM-I        Energy Maximisation Optimisation, Inflexible
EM-F        Energy Maximisation Optimisation, Flexible
CG-I        Continuous Generation optimisation, Inflexible
CG-F        Continuous Generation optimisation, Flexible
HVM        Hourly Variable Market
BM        Baseload Market
CG Days    Continuous Generation Days

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
