# Peer review of "The Potential for Tidal Range Energy Systems to Provide Continuous Power: A UK Case Study"

_jmse, doi:10.3390/jmse8100780_

Round 1

Reviewer 1 Report

1        Title:  Should refer to tidal range energy in the UK

2        Line 24:        Should mention the similar features of tidal stream energy

3        Lines 66/67:  These lines seem to contradict the previous section where timing of production in relation to demand peaks (and price maxima) was identified as the route to maximise profit. 

4        Figure 1:       This must be redrawn using a consistent style of map.

5        Figure 3:       Check legend.  “Left” and “right” do not seem appropriate.

6        Line 429:       Spelling

7        Discussion:   The optimisation has been done over 3 potential plants.  Would including more northerly (e.g. Solway/Morecanbe Bay) or southerly (e.g. Bristol channel/Severn), with presumably further variation in phase shift, offer more attractive optimisation opportunities?

8        Lines 115/116          “hydrodynamic interaction” implies some kind of physical interaction between the schemes affecting water flows.  Is this what is meant?

1        Title:  Should refer to tidal range energy in the UK

2        Line 24:        Should mention the similar features of tidal stream energy

3        Lines 66/67:  These lines seem to contradict the previous section where timing of production in relation to demand peaks (and price maxima) was identified as the route to maximise profit. 

4        Figure 1:       This must be redrawn using a consistent style of map.

5        Figure 3:       Check legend.  “Left” and “right” do not seem appropriate.

6        Line 429:       Spelling

7        Discussion:   The optimisation has been done over 3 potential plants.  Would including more northerly (e.g. Solway/Morecanbe Bay) or southerly (e.g. Bristol channel/Severn), with presumably further variation in phase shift, offer more attractive optimisation opportunities?

8        Lines 115/116          “hydrodynamic interaction” implies some kind of physical interaction between the schemes affecting water flows.  Is this what is meant?

9        Figure 4:       It might be helpful to explain why the great tidal phase difference between the sites (Fig 3) is not reflected in the timing of power generation in Fig 4, where outputs from all schemes seem much more closely in phase.  What characteristics of a scheme would be needed to “fill the gaps” more effectively?

10       Lines 276-280         How realistic is this marketing scenario?  Would it be commercially viable for these three schemes to target the BM? 

11       Lines 429 – 431       I do not understand how tidal stream generators can fill gaps at slack water, when at slack water tidal stream currents at a minimum.

9        Figure 4:       It might be helpful to explain why the great tidal phase difference between the sites (Fig 3) is not reflected in the timing of power generation in Fig 4, where outputs from all schemes seem much more closely in phase.  What characteristics of a scheme would be needed to “fill the gaps” more effectively?

10       Lines 276-280         How realistic is this marketing scenario?  Would it be commercially viable for these three schemes to target the BM? 

11       Lines 429 – 431       I do not understand how tidal stream generators can fill gaps at slack water, when at slack water tidal stream currents at a minimum.

Reviewer 2 Report

A generally well-constructed article which aims at exploring the potential to operate three existing tidal range power plant in the Irish Sea as a single system. I have not found issues which prevent me recommending publication. 

Specific points requiring attention: 
-    There are, here and there, misspelled word or extra word. For example:
line 289: “income by the these two input parameters”
line 397: “on the the daily averaging”
line 429: “by delpoying”-   

Few sentences are very long and confusing. For example:Line 222: “Considering continuous generation this way acknowledges that on certain days the required shift is too high to feasibly (or possibly) cumulatively generate continuously for longer periods”. That is not a major issue but it makes the reading difficult sometimes.       

 -    Figure 2: x-axis title of the bottom right hand side figure: please close the bracket after the symbol ‘degree’.

-    Figure 7-8-9: it is very difficult to distinguish the symbols ‘+’ and the crosses. Perhaps circles and crosses would be a better choice. Moreover, it is probably not necessary to represent every point. The figures are “overloaded”.

 Results:

In the first section you mentioned seven performance indicators. Are they classified following their relevance? If not, could you clearly specify what are the most relevant of them. I did not find this information in the paper.

 Discussion:

The main results of the study do not appear clearly in the discussion. Several scenarios are presented in order to provide continuous power but the best ones are not clearly identified. This is my main concern about this study. I would recommend to revisit the discussion in order to provide the reader with the main results of the study.  

Conclusions:

I would suggest that you revisit the conclusions and bring out the key results of your study.
